# Electrospun Silk Fibroin/*kappa*-Carrageenan Hybrid Nanofibers with Enhanced Osteogenic Properties for Bone Regeneration Applications

**DOI:** 10.3390/biology11050751

**Published:** 2022-05-14

**Authors:** Fahimeh Roshanfar, Saeed Hesaraki, Alireza Dolatshahi-Pirouz

**Affiliations:** 1Biomaterials Group, Department of Nanotechnology and Advanced Materials, Materials and Energy Research Center, Karaj 3177983634, Iran; f.roshanfar@merc.ac.ir; 2Department of Health Technology, Technical University of Denmark, 2800 Kgs. Lyngby, Denmark; aldo@dtu.dk

**Keywords:** silk fibroin, *kappa*-carrageenan, nanofiber, bone regeneration

## Abstract

**Simple Summary:**

Bone tissue engineering has recently been considered as a potential alternative approach to treating patients with bone disorders/defects caused by tumors, trauma, and infection. Scaffolds play a crucial role in the field because they can serve as a template that can provide optimal structural and functional support for cells. In this study, we prepared a series of electrospun silk fibroin/*kappa*-carrageenan nanofibrous membranes with the aim of mimicking bone extracellular matrix structure and composition and improving the biological properties of silk-fibroin-based nanofibers. Our research found that a combinational approach blending *kappa*-carrageenan and silk fibroin could enhance the biological properties of the nanostructured scaffold. *kappa*-carrageenan could also enhance the osteogenic potential and bioactivity properties of silk fibroin nanofibers, while genipin crosslinking preserved the mechanical strength of hybrid nanofibrous mats, indicating that the electrospun hybrid scaffolds could be a potential candidate for bone regeneration applications.

**Abstract:**

In this study, a novel nanofibrous hybrid scaffold based on silk fibroin (SF) and different weight ratios of *kappa*-carrageenan (*k*-CG) (1, 3, and 5 mg of *k*-CG in 1 mL of 12 wt% SF solution) was prepared using electrospinning and genipin (GP) as a crosslinker. The presence of *k*-CG in SF nanofibers was analyzed and confirmed using Fourier transform infrared spectroscopy (FTIR). In addition, X-ray diffraction (XRD) analysis confirmed that GP could cause SF conformation to shift from random coils or α-helices to β-sheets and thereby facilitate a more crystalline and stable structure. The ultimate tensile strength (UTS) and Young’s modulus of the SF mats were enhanced after crosslinking with GP from 3.91 ± 0.2 MPa to 8.50 ± 0.3 MPa and from 9.17 ± 0.3 MPa to 31.2 ± 1.2 MP, respectively. Notably, while the mean fiber diameter, wettability, and biodegradation rate of the SF nanofibers increased with increasing *k*-CG content, a decreasing effect was determined in terms of UTS and Young’s modulus. Additionally, better cell viability and proliferation were observed on hybrid scaffolds with the highest *k*-CG content. Osteogenic differentiation was determined from alkaline phosphatase (ALP) activity and Alizarin Red staining and expression of osteogenic marker genes. To this end, we noticed that *k*-CG enhanced ALP activity, calcium deposition, and expression of osteogenic genes on the hybrid scaffolds. Overall, hybridization of SF and *k*-CG can introduce a promising scaffold for bone regeneration; however, more biological evaluations are required.

## 1. Introduction

Over the last few decades, bone tissue engineering (BTE) has been considered as a potential alternative approach to treating patients with bone disorders/defects caused by tumors, trauma, and infection [1,2,3,4]. Scaffolds play a crucial role in this regard because they can serve as a template that can provide optimal structural and functional support for cells [5]. Functional bone tissue substitutes with a structure and composition similar to native extracellular matrix (ECM) can solve many of the difficulties associated with BTE [6]. Bone ECM is composed of an organic phase, made up of collagen type I and glycosaminoglycans (GAGs), and an inorganic phase, which mainly includes hydroxyapatite and some mineral trace elements such as silicon, lithium, zinc, and magnesium [7]. To date, many biopolymeric materials have been produced to synthesize suitable native-like scaffolds with the ability to support cell migration, proliferation, and differentiation towards new bone tissue formation [8]. Notably, biodegradable natural polymers such as gelatin, silk, collagen, and hyaluronic acid have attracted considerable attention in the field because of their ability to mimic bone’s ECM structure and composition [9].

Among the many biopolymeric materials currently employed for bone scaffolding, *Bombyx mori* silk fibroin (SF) is one of the most commonly used because of its desirable attributes such as biocompatibility, biodegradability, nontoxicity, low immunogenicity, and good mechanical strength [10,11]. SF protein mostly consists of a repeating amino acid sequence (Gly-Ser-Gly-Ala-Gly-Ala)_n_ of hydrophobic nature and is characterized by remarkable mechanical compared to other biopolymers such as alginate, chitosan (CS), and gelatin [10,12]. For instance, in a recent study, it was reported that in a blended membrane consisting of SF/CS/reduced graphene oxide (rGO), the UTS value increased from 1 ± 0.2 (MPa) to 2.8 ± 0.264 (MPa), with increasing SF content. [13]. In another study, an increase in SF content in polyvinyl alcohol (PVA)/SF patches led to an increase in both UTS and tensile modulus by 1.5 and 4 times pure PVA, respectively [14]. SF nanofibers were used to culture osteoblasts, mesenchymal stem cells, and chondrocytes to enhance cell attachment and proliferation [15,16]. However, several studies have shown that when SF is used alone, it cannot support cell differentiation as satisfactorily as other biopolymers such as heparin sulfate and chondroitin sulfate due to the lack of bioactive peptides [17]. Hence, polymer blends of SFs with other bioactive and osteogenic biomaterials have been suggested to enhance its cell/matrix interactions and osteogenesis [18,19].

Carrageenans (CGs) are a family of anionic sulfated polysaccharides, obtained from alkaline extraction of some species of red seaweeds (*Rhodophyta*) [20]. Particularly, *k-*CG has recently gained much more attention in tissue regeneration applications [21]. It is formed of alternate units of β-d-galactose and 3,6-anhydro-α-d-galactose connected by α-(1,3) and β-(1,4) glycosidic unions [20]. *k-*CG has exhibited several capabilities in pharmaceutical fields due to its antiviral, immunomodulatory, anticoagulant, antioxidant, antibacterial, anticancer, and antihyperlipidemic properties, as well as in the field of tissue engineering due to its innate similarity to natural GAGs found in native ECM [20]. Furthermore, the intrinsic thixotropic characteristics of *k-*CG and its strong bonding with proteins and enzymes enable it to be used as an injectable matrix for cell/macromolecule/protein delivery [22]. In our previous study, we produced 3D porous scaffolds from SF and *k-*CG blends, synthesized through freeze-drying. The ability to form an apatite layer on the surface of these scaffolds was investigated by soaking them in simulated body fluid (SBF). Our experimental results revealed that *k-*CG promoted significant biomineralization of composite scaffolds, while there were no signs of apatite formation on SF samples [17].

Following these encouraging results, we tested the electrospinning of SF and *k-*CG polymers in this study. Among several scaffolding techniques, electrospinning has been recognized for its ability to produce nanofibrous mats in a scalable and low-cost manner [23]. Such electrospun mats can mimic the nanotopographies of bone ECM nanofibers (such as collagen) and thereby play a crucial role in cell attachment and proliferation [24]. Indeed, studies have shown that nanofibrous mats are good for cell adhesion due to their higher surface area, which allows better cell adhesion, while their high porosity and pore interconnectivity provide enough space for vascularization required to nourish new bone [25].

Hence, SF/*k-*CG electrospun scaffold materials were developed in this study, in order to mimic the detailed bone ECM structure to enable better cell/scaffold interaction. SF nanofiber fabrication has been studied intensely in the field, but to the best of our knowledge, its blending with *k-*CG has not yet been investigated. We postulate that the blending of *k-*CG and SF could enhance the proliferation/differentiation potential of SF nanofibrous scaffolds. On the other hand, genipin (GP), which is a natural crosslinker was employed, assuming that it could increase the mechanical strength and water-resistant ability of the nanofibrous mats without exhibiting any toxicity symptoms. The morphological aspects of fibers, structural changes, and mechanical properties of hybrid SF/*k-*CG nanofibers were examined and compared with the crosslinked/noncrosslinked pure SF nanofibers using different characterization techniques. Furthermore, in vitro biocompatibility evaluation of the scaffolds, including cell attachment, live/dead staining, MTT bioassay, ALP activity, and Alizarin Red staining, was carried out to evaluate its bioactivity.

## 2. Materials and Methods

### 2.1. Chemicals and Starting Materials

*kappa*-carrageenan, lithium bromide (LiBr), sodium carbonate (Na_2_CO_3_), cellulose dialysis tube (12 KDa, MWCO), ninhydrin, 1,1,1,3,3,3-Hexafluoroisopropanol (HFIP), penicillin, chloroform (CHCl_3_), and [3-(4,-dimethylthiazol-2-y1)-2, 5-diphenyl tetrazolium bromide] were obtained from Sigma-Aldrich (USA), absolute ethanol 99.8%, GP, and isopropanol were provided from Merck Company (Germany). In addition, silk cocoons from *Bombyx mori* were purchased from the Golestan Silkworm Research Institute (Iran). ALP’s kit was obtained from Pars Azmun (Iran). Furthermore, fetal bovine serum (FBS) and other components of cell culture were purchased from Vasegh Company (Iran).

### 2.2. Extraction of SF from Bombyx Mori Cocoons

Purification of SF was performed according to previous studies [17]. To be specific, to remove sericin, *Bombyx mori* was boiled in 0.02 M Na_2_CO_3_ for one hour, then rinsed thoroughly with deionized water three times; finally, it was dried at 37 °C for 24 h. Moreover, degummed SFs were dissolved in 9.3 M LiBr at 80 °C for 8 h. In order to purify SF and eliminate the LiBr from the aqueous solution, dialysis was performed in cellulose tubing against distilled water for three days, changing the water every two hours. Then, the aqueous SF solution was centrifuged to remove impurities, and finally, it was lyophilized until a dried pure SF sponge was obtained.

### 2.3. SF/k-CG Solution Preparation and Co-Electrospinning Process

To obtain 12 wt% SF solutions, the freeze-dried SF sponge was first dissolved in an 8/2 (*v/v*) HFIP/CHCl3 solution and stirred for about two hours using a KA^®^-RCT basic magnetic stirrer. Following that, different amounts of *k*-CG powder (1, 3, 5 mg/mL) were added to the SF solution, and all of the mixing solutions were placed on a shaker plate for 4 h before electrospinning to make a homogenous solution at room temperature. Then, the viscosities of different solutions were measured using a Brookfield viscometer (Model LVDVE 230). Pure SF solution was also considered as the control group, and all selected solutions were poured into a 10 mL syringe with a needle diameter of 0.5 mm for the electrospinning setup (Nanoazma Co., Tehran, Iran). The solution flow rate was fixed to 0.3 mL/h, and droplets of the injected solutions were subjected to a high voltage of 20 kV. Finally, at a distance of 12 cm from the syringe tip, nanofibers were gathered on the aluminum foil surface, then crosslinked with 1% (*w/v*) GP. For this purpose, GP powder was first dissolved in distilled water and agitated for 30 min until a homogenous solution was achieved. Then, the electrospun fibers were immersed in the GP solution for 24 h. The resulting scaffolds are called SF-CG1-GP, SF-CG3-GP, and SF-CG5-GP as GP-crosslinked nanofibers containing *k-*CG with a concentration of 1, 3, and 5 mg/mL, respectively. Further, noncrosslinked and crosslinked pure SF nanofibers were called SF and SF-GP, respectively.

### 2.4. Experimental Characterization

#### 2.4.1. Field Emission-Scanning Electron Microscopy (FE-SEM)

To observe the fiber morphology, high-resolution FE-SEM (TESCAN MIRA3) with a solid-state secondary ion detector (accelerating voltage: 30 keV) was employed. Prior to imaging, the dried samples were cut into round discs (diameter: 5 mm), mounted on an aluminum stub, and sputtered with gold (8–10 nm) for 3 min. The average diameter and fiber distribution were determined by measuring 100 fibers at random from each SEM image and analyzing these images using ImageJ^®^ software (NIH, Bethesda, MD, the USA), as well as the porosity measurements. The results were expressed as “mean ± standard deviation”.

#### 2.4.2. Wettability

For swelling capacity measurements, a certain amount of sample (size of 1 × 1 cm^2^) with an initial weight of $W_{0}$ was soaked in 10 mL of phosphate-buffered saline (PBS) solution at a pH of 7.4 and allowed to completely swell. After 24 h, the sample was removed; filter paper was used to remove the excess PBS on the surface, and the mass of wet sample W was measured. Finally, swelling capacity was calculated using Equation (1).
(1)$$\mathrm{Swelling}\;\mathrm{Capacity}\;\left(\%\right)=\frac{W-W_{0}}{W_{0}}\times 100$$

The wetting test was conducted using a contact angle measuring device (produced at the Materials and Energy Research Center, Iran). First, 4 µL of deionized water was dropped onto the surface of each mat (2 × 2 cm^2^). Then, the images of the droplets were shot with a 2× lens using a DFK 23U618 USB 3.0 color industrial camera. The angles around the image were analyzed, and the average was used to determine the eventual contact angle value.

#### 2.4.3. Fourier Transform Infrared (FTIR) Spectroscopy

Conformational characterization of samples was performed on an FTIR spectrometer (Bruker Vector 33, Ettlingen, Germany). The instrument was purged with nitrogen gas to eliminate the spectral contributions of atmospheric water. Data were collected at a spectral range of 400 cm^−1^ to 4000 cm^−1^ at 4 cm^−1^ resolutions after accumulating 64 scans. To this end, different mats were first prepared in powder form by dipping in liquid nitrogen for five minutes. Then, 0.001 g of each sample was combined with 0.1 g KBr, then molded into discs to carry out necessary tests.

#### 2.4.4. X-Ray Diffraction (XRD) Analysis

The XRD pattern of the samples was evaluated using an automated X-ray diffractometer (Philips PW 3710, the Netherlands). The samples’ scans were performed from diffraction angles of 2θ = 0–40° with a CuKa source (l = 0.154 nm) at a scan rate of 0.015 °/s.

#### 2.4.5. Degree of Crosslinking

A ninhydrin assay was conducted to measure the degree of crosslinking; the percentage of free amino groups in GP-crosslinked nanofibrous samples was compared to noncrosslinked nanofibers. For this purpose, the samples were first heated with a 2 wt% *v/v* ninhydrin solution at 100 °C for 15 min; afterward, the solution’s optical absorbance was measured using a spectrophotometer (Bio-Rad, Hercules, CA, USA) at 570 nm. Glycine solutions of different known concentrations were utilized as standards. Finally, the amount of free amino groups in the mats was proportional to the optical absorbance of the solution. A standard curve of glycine concentration versus absorbance was employed to calculate the concentration of free NH_2_ groups in the sample. The degree of crosslinking was calculated using Equation (2).
(2)$$\mathrm{Degree}\;\mathrm{of}\;\mathrm{crosslinking}=\frac{{\left(\mathrm{NH}\;\mathrm{reactive}\;\mathrm{amine}\right)}_{fresh}-{\left(\mathrm{NH}\;\mathrm{reactive}\;\mathrm{amine}\right)}_{fixed}}{{\left(\mathrm{NH}\;\mathrm{reactive}\;\mathrm{amine}\right)}_{fresh}}\times 100$$
where fresh and fixed are the mole fractions of free NH_2_ remaining in noncrosslinked and crosslinked samples, respectively.

#### 2.4.6. Mechanical Testing

For tensile testing, nanofiber mats were prepared in rectangular pieces with dimensions of 10 × 40 mm^2^ and an average thickness of 0.2 mm based on the method proposed by previous researchers [26,27,28]. Next, the cardboard frame edges were trimmed, and mechanical testing was performed on a universal testing machine (GOTECH AI-3000, Taiwan) at a temperature of 25 °C (humidity: 65%) and an elongation rate of 10 mm/min. The tensile stress–strain curves were processed using the machine-recorded data. The linear portion of the stress–strain curves was used to determine Young’s modulus. Furthermore, ultimate tensile strength (UTS) was calculated by considering the stress at the break to the scaffold’s cross-sectional area.

#### 2.4.7. Mass Loss Measurement

To calculate the biodegradation rate of nanofibrous mats, samples of the same size as those used in swelling tests were separately soaked in 20 mL of PBS (pH: 7.4) containing protease XIV (5 U/mL). At the designated time points of 1, 3, 7, 14, …, 28 days, the samples were removed from PBS, rinsed with deionized water, dried in an oven for 12 h at 37 °C, and weighed. Finally, the mass loss percentage was calculated using Equation (3).
(3)$$Mass\;loss\;\left(\%\right)=\frac{W_{i}-W_{f}}{W_{f}}\times 100$$

The initial and final masses of the nanofibrous mats are represented by $W_{i}$ and $W_{f}$, respectively. All the above-mentioned measurements were repeated three times.

#### 2.4.8. Cell Morphology Assays and Laser Scanning Confocal Microscopy

To study the cell behavior, all nanofibrous mats were clipped into disc-like samples (diameter of 12 mm and height of 0.2 mm) and transmitted into Petri dishes. The samples were sterilized through immersion in 70% ethanol solution for 1 h; then, they were rinsed three times, each for 15 min, in PBS and air-dried in a sterile environment. Human osteoblast-like cells, MG 63, were obtained from the Materials and Energy Research Center (Alborz, Iran). The cells were cultured in Dulbecco’s modified Eagle’s medium (DMEM) supplemented with 10% FBS and 1% penicillin under standard conditions (5% CO_2_, 37 °C) as reported earlier [29]. To observe the initial cell attachment, each sterilized round-shaped nanofibrous mat was seeded with MG 63 cells at a density of 1.0 × 10^4^ cells/well. After one day of cell culture, samples were washed 3 times with PBS solution and fixed in GTA (4% *v*/*v*) for 2 h, then dehydrated with ethanol. Finally, the completely dried samples were covered with a thin gold layer for SEM examination.

The viability of the MG 63 on the scaffolds was also assessed by live/dead staining by acridine orange (Merck; Steinheim, Germany) and propidium iodide (Sigma-Aldrich; Burlington, MA, USA). Cells cultured on the membrane’s surface for 1 and 3 days were followed by removing the culture medium and PBS washing. Then, loaded cell samples were incubated in a mixture of 5 μM acridine orange and 5 μM propidium iodide, in 10 mL culture medium, to assess the presence of both live (green) and dead (red) cells at the same time. After the incubation period (30 min, at room temperature), cell membrane scaffolds were observed using a fluorescence microscope (Olympus CKX53 Microscope; Olympus, Tokyo, Japan).

#### 2.4.9. Evaluation of Cell Proliferation by MTT Bioassay

To assess the eventual cytotoxicity, an indirect 3-(4, 5-dimethylthiazol-2-yl)-2, 5-diphenyltetrazolium-bromide (MTT) bioassay was employed. For this purpose, all samples were extracted in culture medium for 3, 7, and 14 days of incubation at 37 °C. Different extraction solutions were placed in contact with MG 63, with a density of 1 × 10^4^ per well for 24 h. Then, the culture medium was replaced with 10 μL of 12 mM MTT solution and placed in an incubator for 4 h. In the next step, 50 μL of dimethyl sulfoxide (DMSO) was supplied to each well to dissolve the formed formazan crystals. The deposited formazan crystals were dissolved in each well with 50 μL of DMSO, after 10 min of keeping it at 37 °C, the absorbance was measured at 570 nm with an ELISA reader (Stat Fax-2100).

#### 2.4.10. Alkaline Phosphatase (ALP) Activity

The ALP production of MC3T3-E1 cells (osteoblast precursor cell line) seeded on the mats was assessed based on earlier reports [17,30]. Briefly, MC3T3-e1 cells (2 × 10^4^) obtained from the Materials and Energy Research Center (Alborz, Karaj, Iran), were cultured on different samples. Finally, after 3 and 7 days of cell culturing, the cells were lysed using 200 μL of radioimmunoprecipitation lysis buffer system (KalaZist, Tehran, Iran) and 2 μL of phenyl methyl sulfonyl fluoride (PMSF). ALP activity was calculated based on the ALP kit (Vasegh Company, Gorgan, Iran) protocol and was finally normalized corresponding to cell number.

#### 2.4.11. Alizarin Red Staining and Quantification

Alizarin Red staining was used to determine the calcium concentration of the cell membrane scaffold. For this purpose, MC3T3-E1 cells cultured on different samples after 14 and 21 days in osteogenic medium (DMEM, 0.1 µg/mL dexamethasone, 50 µg/mL ascorbic acid, and 10 Mm β-glycerophosphate) were fixed with 4% paraformaldehyde in 20 min, then washed with PBS. A 0.5 g amount of Alizarin Red-S was dissolved in 25 mL of deionized water, and the pH was adjusted to 4.2 using 0.5% ammonium hydroxide. Each well received one milliliter of Alizarin Red-S solution, which was then incubated for one hour at room temperature. Subsequently, the samples were properly rinsed with deionized water, then dried at ambient temperature before being examined under an inverted microscope (Olympus CKX53 Microscope, Olympus, Tokyo, Japan). To quantify the calcium concentration, each stained sample was moved into a fresh 24-well plate and treated with 1 mL of 10% cetylpyridinium chloride solution for one hour to desorb calcium ions. The solution’s absorbance was measured at 540 nm in an ELISA reader (Stat Fax-2100), and the results were normalized by cell number.

#### 2.4.12. Gene Expression Analysis

The expression of some osteogenic genes, including collagen type I (Col I) and run-related transcription factor 2 (Runx2), was performed using the real-time PCR technique. The MC3T3-E1 cells were seeded at a density of 5 × 10^5^ cells per sample (disc-like sample with a diameter of 5 mm and thickness of 2 mm). After 24 h of attachment, the culture medium was replaced with 1 mL of osteogenic medium (DMEM, 0.1 µg/mL dexamethasone, 50 µg/mL ascorbic acid, and 10 Mm β-glycerophosphate). The osteogenic medium was changed every 2 days until the end of the experiment. After 7 days of cell culture on the scaffolds, the total RNA was extracted using the Qiagen RNeasy Mini kit (Qiagen, South Korea) according to the manufacturer’s instructions, then converted to complementary DNA (cDNA) with first-strand cDNA using the TaKaRa RNA PCR kit, Shiga, Japan. The differentiation of MC3T3-E1 was monitored by measuring mRNA expression levels of Col I and Runx2. All the fold changes in gene expression were normalized to GAPDH. All results were quantified using the 2^− (ΔΔCT)^ method. The genes and the related specific primer sequences for Col I and Runx2 were designed and illustrated in Table 1 [31]. Each measurement was assessed in triplicate.

### 2.5. Statistical Analysis

All of the experiments were carried out at least three times, and the findings were averaged and given as mean ± SD. A statistical significance threshold of *p* < 0.05 was used in the statistical calculations.

## 3. Results and Discussion

### 3.1. Fiber Characterization

The morphology of SF and SF/*k-*CG hybrid nanofibers was observed with SEM. The average fiber diameter, porosity percentages, and viscosity of the electrospun solution are presented in Table 2. The morphology and size distribution of noncrosslinked SF fibers, crosslinked SF fibers (SF-GP), and crosslinked-hybrid fibers (SF-CG1-GP, SF-CG3-GP, and SF-CG5-GP) are also given in Figure 1a. Evidently, all samples display a homogeneous, continuous, and bead-free nanofibrous structure. Moreover, individual noncrosslinked SF fibers had an average diameter of 376.7 ± 7 nm, while crosslinking with GP caused a slight increase in the SF nanofibers’ diameter to 390 ± 5 nm, thus resulting in greater interaction among the polymer chains. Additionally, we could see that the nanofiber diameter increased with an increase in CG content to 410 ± 2 nm for SF-CG1-GP, 450 ± 3 nm for SF-CG3-GP, and 501 ± 6 nm for the SF-CG5-GP sample. This may be associated with the CG viscosity that in return increases the electrospun solution viscosity and thus fiber diameter. These findings are in good agreement with those obtained by Zadegan et al. [32], who reported that the incorporation of viscous nettle into electrospun SF mats could increase the diameter of the nanofibers, causing polymer chain entanglement.

### 3.2. Wettability

Scaffold wettability plays a significant role in tissue regeneration, as a high water swelling ratio is conducive to transferring nutrition and wastes [33]. Hence, the swelling capacity of samples was measured and is depicted in Figure 1b. The crosslinked SF-GP mat has a lower swelling capacity at the equilibrium point of 131 ± 0.93% than the noncrosslinked SF mat (150 ± 1.18%). As expected, increasing *k-*CG also increased the swelling capacity to 160 ± 1.5%, 170 ± 1.8%, and 190 ± 1.98% for SF-CG1-GP, SF-CG3-GP, and SF-CG5-GP, respectively. We believe this to be related to the hydrophilic nature of *k-*CG due to its abundant hydroxyl groups [17]. The contact angle data showed similar trends to those observed in Figure 1b. As noted in Figure 1b, hybrid mats showed a decrease in contact angles with increasing *k-*CG content. SF-CG5-GP nanofibers had the lowest water contact angle (θ = 46.4 ± 3.7°), while these values were 57.1 ± 1.1, 62.5 ± 1.3, 83.8 ± 0.1, and 83.2 ± 0.2 for F-CG3-GP, F-CG1-GP, SF-GP, and SF respectively. It was therefore concluded that *k-*CG, due to its hydrophilic nature, results in more suitable wettability. It has been reported that surfaces with a water contact angle in the range of 40–70° are optimal for osteoblast attachment and proliferation [34,35]. The presence of hydrophilic *k-*CG in nanofibrous mats can effectively moderate the hydrophobic properties of SF-based nanofibers, which are mostly generated by SF amino acid domains such as glycine-alanine and glycine-serine [36].

### 3.3. FTIR

FTIR analysis was performed to examine the functional groups contained in each membrane scaffold, and the results are presented in Figure 1c. The spectrum of the noncrosslinked samples (SF) shows bands at around 713 cm^−1^ for amide v, 1235 cm^−1^ for amide III, 1533 cm^−1^ for amide II, and 1673 cm^−1^ for amide I in the SF structure. After GP crosslinking, the bands corresponding to amide III, amide II, and amide I slightly shifted to 1260, 1524, and 1640 cm^−1^, respectively. This could be due to the change in conformation from random coils or α-helices to β-sheets reminiscent of a more stable SF structure [37]. Some characteristic bands of GP at 1443 (–CH3, CH2) and 1105 (–COH) cm^−1^ appeared as well [38]. Moreover, a new characteristic peak appeared at 1160 cm^−1^, which could be associated with the stretching vibration peak of tertiary amine C-N as a result of the emerging covalent bond in the SF-GP sample, which was due to the covalent crosslinking between GP and amino groups in SF [39]. Genipin’s crosslinking mechanism is complicated and is not exactly understood; however, it is recognized that it uses a ring-opening reaction to crosslink materials containing primary amine groups. This reaction can be initiated by free amino groups via nucleophilic attack against the olefinic carbon atom at C-3 on the GP heterocyclic ring. This eventually leads to the dihydropyran ring of GP forming a new aldehyde group. Subsequently, the newly formed aldehyde reacts with another amine group of SF, and the resulting covalent conjugation is formed [37,38].

In hybrid nanofibrous scaffolds (SF-CG1-GP, SFCG3-GP, and SFCG5-GP), the bands that appear around 1265 and 1050 cm^−1^ are related to the sulfate group (stretching S-O) and galactopyranose rings (C-C vibrations) of k-CG, respectively. With increasing k-CG content, sharper sulfate-related peaks were observed. Additionally, we found a peak at around 2910 cm^−1^ attributed to C-H bonds in both SF and k-CG. Peaks at around 3400 cm^−1^ are attributed to hydrogen bonds (OH groups) that can occur between nanofiber components [17]; in the presence of GP, this peak becomes sharper due to reaction with GP. The FTIR results confirm the following: (1) the successful incorporation of GP into the nanofiber structure; (2) the formation of hydrogen bonding (between GP, SF, and CG); (3) the formation of covalent bonds (between GP and SF).

### 3.4. XRD Patterns

As shown in Figure 1d, no obvious 2θ peak was observed in SF samples, indicating that the SF conformation mainly consisted of random coils. However, SF-GP samples exhibited peaks at 2θ = 20°. Moreover, a new peak at 2θ = 24° appeared, and its intensity increased with increasing *k-*CG content. The peak is characteristic of the β-sheet crystalline structure, and it is suggested that after GP crosslinking, the amorphous SF (Silk I) is partially transformed into a more rigid and stable β-sheet structure (Silk II). This finding proves that GP could be acting as a crystallization-inducing agent. The other hybrid scaffolds exhibit a similar pattern to crosslinked SF nanofibers, thereby indicating that the presence of *k-*CG in the nanofiber composition does not damage their crystalline structure. This result is consistent with the FTIR results, demonstrating that the crosslinked nanofibers had a higher percentage of β-sheets due to the addition of GP, which caused SF molecules to be structurally rearranged. These findings were also consistent with the results of a previous study by Wang et al. reporting that the uncrosslinked SF film showed arc-shaped scattering at around 20°, corresponding to the amorphous silk (Silk I), while after GP crosslinking, there appeared an obvious characteristic peak at around 20.7° and 24.3°, illustrating the content of β-sheet (Silk II) conformation [40].

### 3.5. Degree of Crosslinking

The crosslinking degrees of different fabricated mats were assessed using the ninhydrin assay. The obtained values were 65.2 ± 1%, 55 ± 0.5%, 49.5 ± 2.5%, and 42.3 ± 1.1%, for SF-GP, SF-CG1-GP, SF-CG3-GP, and SF-CG5-GP, respectively. Since GP reacts with the SF amino acids spontaneously, the degree of crosslinking increases with increasing SF content. The maximum percentage (65.2 ± 1%) is related to SF-GP, which contains the highest concentration of amino groups in its structure.

### 3.6. Mechanical Strength

The results of the mechanical studies are shown in Figure 2a–c. Figure 2a shows the stress–strain curves for pure SF, SF-GP, and hybrid fibers. According to Figure 2b, crosslinking with GP increases the UTS in the SF-GP group (UTS = 8.50 ± 0.3 MPa) by approximately 100%, compared to the noncrosslinked SF mat (UTS = 3.91 ± 0.2 MPa). This could be due to the formation of covalent bonds between GP and SF, which was confirmed by the FTIR analysis. Furthermore, the addition of GP increased the percentage of β-sheets in the crosslinked nanofibers, thus leading to stiffer and more robust nanofiber mats. This outcome is consistent with those of previous studies in which GP-crosslinked samples had higher mechanical strength than other scaffolds [39,41]. As expected, with increasing *k-*CG content, the UTS was reduced to 7.31 ± 0.2 MPa in SF-CG1-GP, 5.20 ± 0.2 MPa in SF-CG3-GP, and 4.04 ± 0.09 MPa in SF-CG5-GP. These results are in agreement with the SEM analysis, which established a correlation between increasing *k-*CG content and a larger fiber diameter. Several studies have already proved that the reduction of fiber diameters enhanced tensile strength because of the sharper orientation of the polymeric chain along the fiber axis [42,43]. The higher tensile strength of the SF-GP sample could be associated with the increased SF content compared to hybrid mats, since the β-sheet structure in SF increased its tensile strength [39,44]. Previous studies have demonstrated that with increasing SF content, the compressive and tensile moduli of the samples increased significantly due to the rigid β-sheet structure of SF [13,45]. Another important factor in mechanical strength is the porosity percentage of the scaffolds [17,46]. According to Table 1, the sample SF-GP with the lowest porosity has logically the highest UTS.

A similar trend was also obtained for Young’s modulus, as shown in Figure 2c (SF, 9.17 ± 0.3 MPa; SF-GP, 31.2 ± 1.2 Mpa; SF-CG1-GP, 28.7 ± 0.9 Mpa; SF-CG3-GP, 25.4 ± 0.3 Mpa; and SF-CG5-GP, 18.9 ± 0.5 Mpa). These findings suggest that the SF-based electrospun samples enjoy similar mechanical properties to non-weight-bearing bones, and they could be viable candidates for bone tissue applications [13].

### 3.7. Mass Loss Measurement

The characterization of the degradation behavior of nanofibers is shown in Figure 2d. The noncrosslinked SF fiber mats had the highest degradation rate by about 35% after 28 days of incubation in protease XIV solution. After GP crosslinking, the SF macromolecular chains were linked via covalent bonds, making the nanofibers more stable (SF-GP: mass loss: 10.2%); this stability could be also partially due to the crystalline transition of SF from random coils to β-sheets during GP crosslinking [13,39,45]. The degradation rates for SF-CG1-GP, SF-CG3-GP, and SF-CG5-GP mats increased to 18.5, 21, and 29%, respectively, after 28 days of incubation in protease XIV solution. Such a boost in weight loss percentage could be associated with more hydrophilicity properties of *k-*CG and the amorphous structure, which can be susceptible regions for enzymatic attack [20]; this finding is also verified by our swelling capacity assessment.

### 3.8. Cellular Morphology and Viability

The morphology of osteoblast-like cells MG 63 after 24 h of cells seeding on the surface of different mats is illustrated in Figure 3a. SEM evaluation of nanofiber mats revealed that the cells were well attached on the surface of all samples, especially on the hybrid samples, by their lamellipodia and small pseudopods. In fact, a flat morphology of cells and good cell spreading on scaffolds’ surfaces were observed, especially for the SF-CG5-GP sample containing the most *k-*CG content. In comparison to SF nanofibrous mat, where the cell phenotype is rounded, on hybrid samples, cells were shown to be elongated and adopted a spindle-shaped morphology, showing that *k-*CG has a favorable influence on cell attachment to the matrix. In fact, based on the findings of many other studies, hydrophilic and nanorough surfaces such as *k-*CG-containing materials promote the adsorption of a wide variety of glycoproteins such as vitronectin and fibronectin, consequently promoting the adhesion and spreading of osteoblast or osteoblast-like cells [17,47].

The distributions and viability of MG 63 cells in electrospun nanofibers were visualized using confocal microscopy following live/dead cell staining. As shown in Figure 3b,c, all cells were alive (stained green) in all nanofibrous membranes, after 1 and 3 days of cell culture, and there were no dead cells (stained red) found. The variation in the number of live cells and cell spreading is because of the nanofiber composition. More live cells could be observed on SF-CG5-GP with the highest carrageenan content, and the cells spread and distributed on it more evenly than pure SF nanofibers.

### 3.9. Cell Proliferation

Cytotoxicity of nanofibers against the MG 63 cell was determined by an MTT bioassay and is presented in Figure 4a. To evaluate the proliferation rate of MG 63 on the different nanofiber mats, the number of viable cells after 4 h of cell culture was counted and utilized as a reference point. The biocompatibility of mats was confirmed, as all samples had more than 90% cellular viability during the 14 days of culturing. Moreover, no trace of toxicity in MG 63 cells was observed after GP crosslinking. Although the cell number increased with culture time, there was a difference in cell proliferation rate depending on nanofiber composition. Generally speaking, the samples containing *k-*CG exhibited better cell viability than the pure SF nanofibers. More cellular proliferation was observed on the sample with the highest *k-*CG content (SF-CG5-GP). On Days 7 and 14, significant differences (*p* < 0.001, *p* < 0.01) in cell number were observed between SF-CG5-GP and SF nanofibers. Improved cell proliferation can be associated with the presence of *k-*CG, which improves the surface hydrophilicity of hybrid fibers, resulting in greater absorption of bioactive proteins [48]. It has been reported that the sulfated disaccharide-saturated structure of *k-*CG and its negatively charged functional groups, similar to native ECM characteristics, could provide a more suitable matrix for adhesion and proliferation of osteoblastic cells [48,49].

### 3.10. ALP Activity

ALP is a biomarker of osteogenesis that appears in the early stage of osteogenic differentiation and is linked to osteoblast cell mineralization activity [50,51]. Calcification occurs through the hydrolysis of phosphate esters by ALP, which leads to the accumulation of inorganic phosphates and Ca^2+^ and thereby increasing ECM mineralization. The ALP enzyme is a transcription factor that activates the cell differentiation factors, as well as playing a role in message transmission and osteoblast differentiation [52]. ALP activity of MC3T3-E1 cultured in nanofibrous scaffolds was assessed by using an ALP kit, and the results are shown in Figure 4b. According to Figure 4b, the ALP levels of mats containing *k-*CG were significantly higher than that of pure SF, with the rate increasing over time. The highest amount of ALP expression was observed on SF-CG5-GP on Day 7. This could be attributed to the presence of sulfonate groups in *k-*CG. A similar trend was observed by Felgueiras et al., who found that grafting of sulfonate groups on Ti alloys could improve osteoblast differentiation [53].

### 3.11. Mineralization of MC3T3-E1

Figure 4e displays the development of mineralized ECM, a late osteogenic marker, on the different nanofibrous mats on Days 14 and 21. The red dots are evident from these images, indicating the formation of Alizarin Red/calcium complexes. These were higher in the hybrid nanofibers than pure SF in both incubation times. In addition, Figure 4c shows that the calcium deposits produced by MC3T3-E1 on SF-CG5-GP were significantly higher than those of SF and SF-GP on Day 21 (*p* < 0.01). This is in accordance with the previously suggested action of CG on promoting osteogenic differentiation [48]. In fact, the existence of sulfonate groups (-SO_3_H) in CG causes Ca^2+^ to be taken up, followed by phosphate ions, resulting in calcification via nucleation and growth. In mineralized ECM, Alizarin Red-S can bind to Ca^2+^, leading to bright red stains [16].

### 3.12. Expression of Osteogenic Markers

The osteogenic ability of pure SF, SF-GP, and SF/*k-*CG hybrid nanofibrous scaffolds was evaluated through a quantitative analysis of relative mRNA expression levels of the main osteogenic genes, including Col I and Runx2. Col I is the most abundant protein found in the organic phase of the bone ECM. Collagenous matrix formation is also important for the development of bone cell phenotype and is thought to be a cell differentiation modulator. The relative gene expression of Col I from seeded MC3T3-E1 under osteogenic conditions is presented in Figure 4d. A very significant upregulation was observed in cultured cells on SF-CG5-GP scaffolds compared to pure SF and SF-GP samples. This behavior reiterates the fact that *k-*CG enhances osteogenic differentiation of MC3T3-E1 cells. Runx2 is also an important transcription factor that activates the osteoblast differentiation marker genes in all osteogenic differentiation processes. Both Col I and Runx2 genes exhibited low expression levels in the cells on the pure SF and SF-GP scaffolds while showing the highest expression on the SF-CG5-GP nanofibers. From the literature, the bioactive sulfonate groups of *k-*CG were reported to be associated with an increase in the expression of osteogenic marker genes. It was reported that *k-*CG could promote MC3T3-E1 preosteoblast adhesion and spreading, metabolic activity, proliferation, and osteogenic differentiation [54].

## 4. Conclusions

In this study, we prepared a series of electrospun SF/*k-*CG nanofibrous membranes with the aim of mimicking bone ECM structure and composition and improving the biological properties of SF-based nanofibers. It was concluded that a combinational approach blending *k-*CG and SF could enhance the biological properties of the nanostructured scaffold. *k-*CG could also enhance the osteogenic potential and bioactivity properties of SF nanofibers. while GP crosslinking preserved the mechanical strength of hybrid nanofibrous mats. The incorporation of *k-*CG into the SF nanofibrous scaffolds in electrospun nanofibers was confirmed by FTIR, XRD, and contact angle measurements. Wettability studies indicated that the presence of *k-*CG in nanofibrous mats could effectively moderate the hydrophobic properties of SF-based nanofibers, which in turn enhanced cell viability and proliferation. On the other hand, degradation studies proved that the GP crosslinking process could stabilize SF nanofibrous scaffolds and make them more stable in biological solutions. Furthermore, GP crosslinking increased the mechanical characteristics of the scaffolds, which is another crucial aspect of bone tissue engineering. From the regulation of ALP activity, mineral deposition, and osteogenic gene expression, *k-*CG was found to guide MC3T3-E1 toward the osteogenic lineage and induce the mineralization and formation of bone tissue in vitro. In particular, SF-CG5-GP displayed the highest effects on MC3T3-E1 in terms of ALP activity, calcium deposition, and Col I and Runx2 gene expression, indicating that it could be a potential candidate for bone regeneration applications, though more in vitro and in vivo studies are inevitable.

## Figures and Tables

**Figure 1 biology-11-00751-f001:**
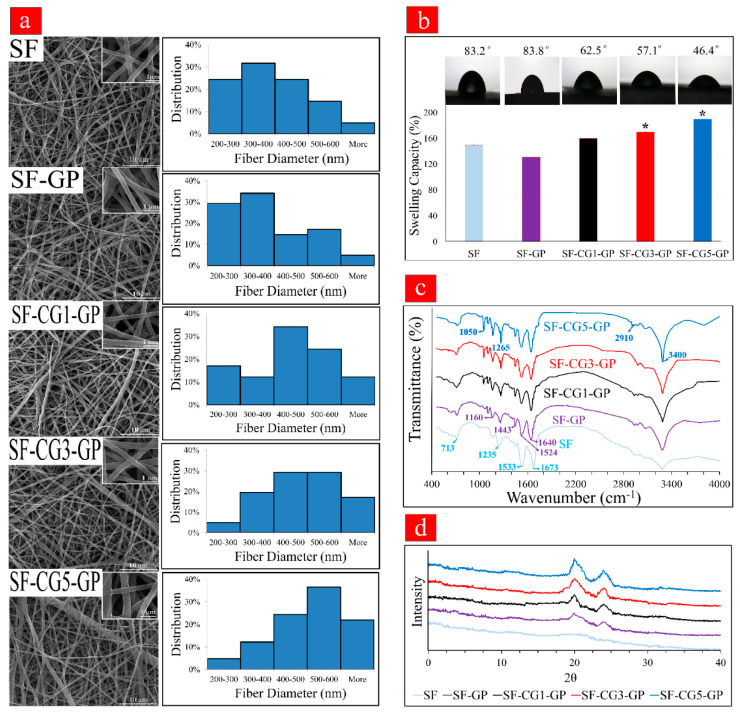
(**a**) SEM images and the fiber diameter distribution of all electrospun mats. (**b**) Wettability of different electrospun mats in terms of swelling capacity and contact angle. (* shows significant differences between each group and SF at *p* < 0.05). (**c**) FTIR spectra of all nanofibrous scaffolds. (**d**) XRD patterns of different samples.

**Figure 2 biology-11-00751-f002:**
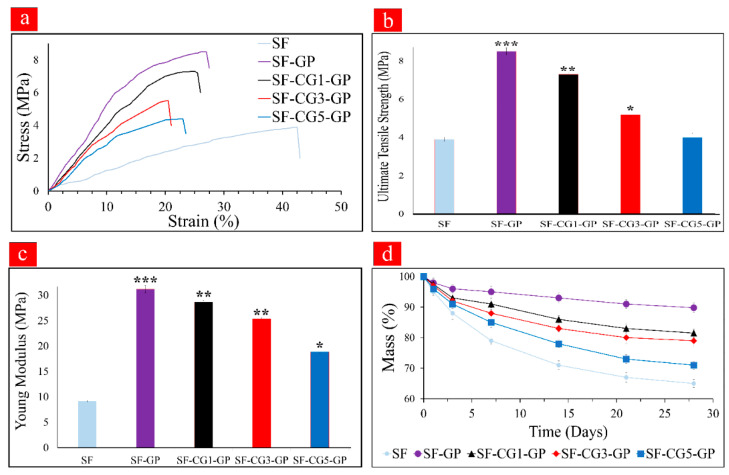
(**a**) Stress–strain curves, (**b**) UTS, and (**c**) Young’s modulus of nanofibrous mats. (***, ** and * show significant differences between each group and SF at *p* < 0.001, *p* < 0.01 and *p* < 0.05 respectively). (**d**) Degradation behavior of nanofibrous mats in protease XIV solution.

**Figure 3 biology-11-00751-f003:**
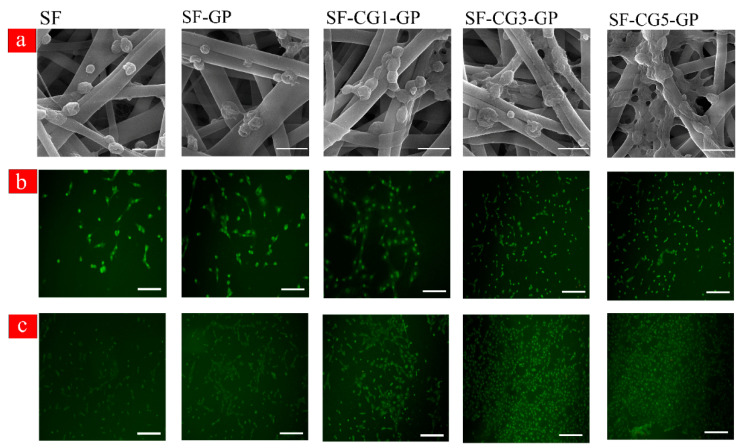
(**a**) SEM images of the initial attachment of MG 63 cells after 24 h of culturing them on the samples. Bar is 1 µm. (**b**) Live-dead staining of MG 63 cultured in electrospun nanofibrous scaffold on Day 1 and (**c**) on Day 3. Bar is 100 µm.

**Figure 4 biology-11-00751-f004:**
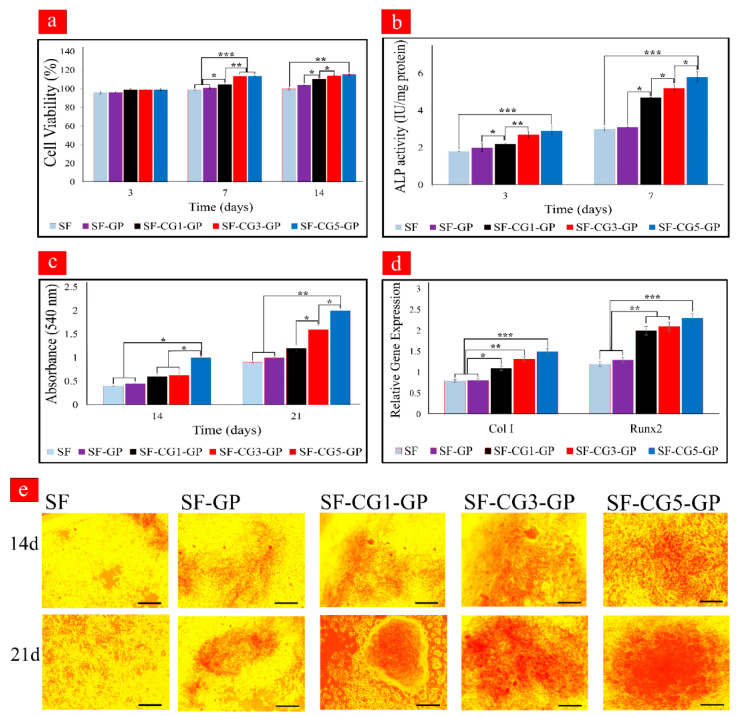
(**a**) Proliferation of MG 63 cells in electrospun nanofibrous scaffolds by MTT. (**b**) ALP activity of MC3T3-E1 cells cultured on the nanofibrous mats. (**c**) Quantitative evaluation of mineral deposition of calcium content of MC3T3-E1 seeded on electrospun scaffolds. (**d**) Relative Gene Expression of Col I and Runx2. (**e**) Alizarin Red staining of MC3T3-E1 cultured on electrospun scaffolds on Days 14 and 21. Bar is 200 µm (*** *p* < 0.001, ** *p* < 0.01, * *p* < 0.05).

**Table 1 biology-11-00751-t001:** Primer sequence for real-time PCR.

Gene Name	Forward/Reverse
RUNX2 COLI GAPDH	GCCTCCAAGGTGGTAGCCC/CGTTACCCGCCATGAGAGTA TCCGACCTCTCTCCTCTGAA/GAGTGGGGTTATGGTGGGAT ACAGTCAGCCGCATCTTCTT/ACGACCAAATCCGTTGACTC

**Table 2 biology-11-00751-t002:** Properties of different samples.

Sample	Viscosity of Electrospun Solution (mPa.S)	Average Fiber Diameter (nm)	Porosity (%)
SF	438 ± 2	376 ± 7	49.9 ± 1.20
SF-GP	470 ± 9	390 ± 5	40.5 ± 3.01
SF-CG1-GP	533 ± 2	410 ± 2	53.2 ± 4.21
SF-CG3-GP	580 ± 7	450 ± 3	57.5 ± 5.1
SF-CG5-GP	617 ± 2	501 ± 6	61.5 ± 3.3

## Data Availability

Not applicable.

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
