# Peer review of "Electrospun Silk Fibroin/*kappa*-Carrageenan Hybrid Nanofibers with Enhanced Osteogenic Properties for Bone Regeneration Applications"

_biology, 2022, doi:10.3390/biology11050751_

Round 1

Reviewer 1 Report

In this study, the authors propose a nanofibrous hybrid scaffold based on silk fibroin (SF) and different weight ratios of kappa-carrageenan (k-CG), as a novel potential plataform for bone regeneration applications. Both structural and biological characterization of the scaffolds are investigated. The manuscript is interesting and could open new perspectives in the field of tissue engineering, but the authors should better describe the real advantages introduced by the scaffold proposed. Biological studies need to be better presented and require some insight. In particular, it is unclear whether differentiation medium was used for all investigations or only for mineralization results. For completeness, the differentiation study should include the expression analysis of some key osteogenic mRNAs (osteopontin, osteocalcin, RUNX2, etc.).

Author Response

Reviewer 1:

Comment #1:

In this study, the authors propose a nanofibrous hybrid scaffold based on silk fibroin (SF) and different weight ratios of kappa-carrageenan (k-CG), as a novel potential plataform for bone regeneration applications. Both structural and biological characterization of the scaffolds are investigated. The manuscript is interesting and could open new perspectives in the field of tissue engineering.

Author response #1:

We would like to thank the referee very much for his/her positive feedbacks.

Comment #2:

The manuscript is interesting and could open new perspectives in the field of tissue engineering, but the authors should better describe the real advantages introduced by the scaffold proposed.

Author response #2:

Thank you for your valuable feedback! In response to the respected referee's concerns, we revised the manuscript, particularly the introduction and conclusion sections.

Comment #3:

Biological studies need to be better presented and require some insight. In particular, it is unclear whether differentiation medium was used for all investigations or only for mineralization results.

Author response #3:

Thank you for mentioning this concern! In the revised manuscript, we clarified the types of culture medium which were used in different in-vitro assays. Actually we used  Dulbecco's Modification of Eagles medium (DMEM) supplemented with 10% FBS and 1% penicillin for Cellular Morphology and Viability and Cell Proliferation assays; and osteogenic medium (DMEM, 0.1 µg/mL dexamethasone, 50 µg/mL ascorbic acid, and 10 Mm β-glycerophosphate) just was used for the mineralization and gene expression analysis. We added this information in more details, in the manuscript.

Comment #4:

For completeness, the differentiation study should include the expression analysis of some key osteogenic mRNAs (osteopontin, osteocalcin, RUNX2, etc.).

Author response #4:

We have provided our data for gene expression analysis and added this section to the manuscript (Figure 4-d).

Reviewer 2 Report

Please check the attached document.

Author Response

Reviewer 2:

Comment #1:

The manuscript by Roshanfar et al. describes the preparation and characterisation of electrospun (bombyx mori) silk fibroin-kappa carrageenan composite microfibre mats, with a view to their potential use in bone regeneration applications. In my opinion, this manuscript reports the wide ranging work rather superficially, raising several unanswered questions, which I suggest should be discussed in greater detail in a revised manuscript.

Author response #1:

According to the respected referee’s concerns, we made required/suggested changes in the manuscript and we worked diligently to address the reviewer comments.

Comment #2:

Page 2 and P8: The authors state that the main repeating sequence of the fibroin (-Gly-Ser-Gly-Ala- Gly-Ala-) is hydrophobic. It may be surprising that chain segments containing these relatively small amino acids would be hydrophobic, in view of the extensive hydrogen bonding capacity of the peptide linkages and the hydroxylated Ser side-group. The authors should at least provide a reference to support that claim.

Author response #2:

In reference number "12" which is cited in our manuscript, it was clearly mentioned that the primary structure of SF mainly comprises repetitive blocks of hydrophobic heavy chains (H-fibroin, Mw = 391.6 kDa). Hydrophobic H-fibroin consists of 45.9% Gly, 30.3% Ala, 12.1% Ser,), which its sequence can be described as (–Gly–Ser–Gly–Ala–Gly–Ala–)n.

Comment #3:

P3: The authors state that 'Some chemicals including k-CG, lithium bromide (LiBr),...were obtained from Sigma-Aldrich (USA),' This phrasing is rather confusing, as it suggests that other chemicals were used, but their origin is not discosed. I suggest it would be better to start the sentence: 'Kappa- carrageenan, lithium bromide (LiBr),..'

Author response #3:

Thank you for mentioning this point! You are right! We revised the manuscript based on your suggestion.

Comment #4:

P3: The authors describe 'a 10 ml syringe with an inner diameter of 0.55 mm for electrospinning'. Are these figures correct? If so, I estimate the syringe would be over 4 m in length. Does 0.5mm i.d. refer to a needle attached to the syringe? Please clarify this.

Author response #4:

We meant "a needle diameter of 0.5 mm". We have corrected it in the manuscript.

Comment #5:

P3: Further details of the FE-SEM method are required (e.g. accelerating voltage, how specimens were mounted, Au thickness or coating conditions).

Author response #5:

In the FE-SEM method section (page 4), we added the required details including: accelerating voltage, Au thickness and how specimens were mounted.

Comment #6:

P4: The authors state that '...a filter paper was used to remove the excess water on the surface...' Presumably the liquid was PBS solution - i.e. not water.

Author response #6:

We replaced the "PBS", in page 4.

Comment #7:

P4: More details are required for the FTIR method, please. What was the wavenumber resolution; how many scans were collected; was the optical path purged to exclude atmospheric water and carbon dioxide?

Author response #7:

More required details are provided in FTIR method section, in page 4.

Comment #8:

P4: It is not clear how genipin crosslinking is supposed to work with silk fibroin. Native silk fibroin contains relatively few -NH2 groups, associated with Lys (about 0.3% molar of overall amino acids) - or at the N-terminus. If crosslinking depends on -NH2 at the N-terminus, its concentration will be affected by fibroin depolymerisation, which happens during the dissolution process.

Is the delocalised -NH2 in Arg sufficiently reactive to be involved? Other >NH groups occur as amides (i.e. in peptide linkages, and in Asn or Gln), or as part of an aromatic heterocycle (His, about 0.2% molar of amino acids); can these react?

Can the authors comment, please, or provide some appropriate references?

Author response #8:

Although, the Genipin crosslinking mechanism is complicated and is not exactly understood, in a previous study (reference 41) it has reported that the free amino groups (mainly the free amino group of the molecular chain end group and e-NH2 on the amino acid such as lysine) were suitable and enough in the SF, which could onset nucleophilic attack to the 3-C on the genipin six-membered heterocyclic ring, directly caused the dihydropyran ring of genipin to form a new aldehyde group and a secondary amine. Following formation-elimination reaction occurs between the secondary amine and the aldehyde group. Ultimately, a new covalent bond is formed; a tertiary amine on a molecule of genipin connects the SF. The genipin cross linking mechanism is discussed in page 9, and we provided more appropriate references for this section.

Comment #9:

P5: Was excess liquid removed prior to drying for mass loss measurements? Please specify. Otherwise, the weights recorded may be erroneously high due to retained PBS salts.

Author response #9:

Yes, prior to drying, the samples were rinsed with deionized water to remove PBS salts. We added this detail to the Mass Loss Measurement section in the manuscript, in page 5.

Comment #10:

P6: It is stated that the alizarin red was dissolved in deionised water at pH 4.2. How was the pH of 4.2 achieved for deionised water? E.g. was there a buffer dissolved in deionised water, or was the deionising apparatus faulty, allowing some acidic species to pass?

Author response #10:

Actually, pH was adjusted to 4.2 using 0.5% ammonium hydroxide. We added this detail to the manuscript, in page 6.

Comment #11:

P6 and Table 1: the authors mention solution viscosity and its putative effects on fibre spinning. If viscosity might be so important, the authors should describe how it was measured and discuss the results in more detail, please.

Author response #11:

Since solution viscosity affects on the electrospun fiber diameter, we measured viscosities of different solutions using a Brookfield viscometer. We added more details about it in the manuscript.

Comment #12:

P6 onwards: Can the authors provide a more detailed discussion of the IR spectra, please? E.g. are there features in the spectra related to the amounts of carrageenan added?

Author response #12:

Yes, as stated in the manuscript, upon increasing k-CG content, sharper sulfate-related peaks were observed.

Comment #13:

An IR band around 1644 cm-1 could be due to the amide I band of the silk protein, rather than double bonds due to reaction with genipin. Can the authors comment and provide references explaining what structures correspond to the IR wavenumbers, please?

Author response #13:

In a previous study (reference 17) it was reported that peak appeared at 1636 is related to the imine conjugation between the amino groups in SF and aldehyde groups in glutaraldehyde; so based on the cited references, the authors believe that band around 1644 cm-1 is more likely to be related to imine conjugation between the amino groups in SF and aldehyde groups in Genipin. The shoulder at 1693 can be related to the amide 1 bonds in SF protein.

Comment #14:

NH2 groups are relatively scarce in silk fibroin. Can the authors comment on what concentrations are expected to correspond to the relatively large IR peak observed, please?

Author response #14:

Thank you for your comment! We dissolved the freeze-dried SF sponge to obtain 12 wt% SF solutions.

Comment #15:

The amide I region (1600 to 1700 cm-1, in Fig. 1c) of the SF specimen looks strange, with what appear to be two strong bands. Can the authors explain this, please?

Author response #15:

Based on the cited references, the two bond appeared between 1600 to 1700 cm-1 are related to:1) imine conjugation (C=N) which formed between the amino groups in SF and the aldehyde groups produced in GP (peak at 1644 cm−1), 2) the C=C double-bond ring and the stretching modes of the core of the GP molecules (the absorption band at 1628 cm-1).

Comment #16:

Also, it is quite difficult to see the important information, particularly in the IR. Since the main information is contained within a limited part of the wavenumber range, it would be very useful to provide an additional figure focused on these regions.

Author response #16:

In the revised manuscript, the authors tried to insert some important information (specific peaks) in the FTIR figure to be seen well. We also worked on the resolutions.

Comment #17:

Figs. 1 and 2: It would be very useful to add tick-marks to the x-axes, please.

Author response #17:

We added tick-marks to the x-axes of the figures.

Comment #18:

P8: How do the XRD patterns shown compare with previously published data for silk I and II? Can the authors comment and provide appropriate references, please?

Author response #18:

The obtained XRD results were consistent with the results of a previous study (reference 43) reporting that the uncrosslinked SF film showed arc-shaped scattering at around 20°, corresponding to the amorphous Silk (silk I), while after GP-crossliking they appeared an obvious characteristic peak at around 20.7° and 24.3°, illustrating  the content of β-sheet(silk II) conformation. Appropriate references are cited in the manuscript.

Comment #19:

P9: What do the degrees of crosslinking indicate? For example, can the authors estimate the numbers of links between fibroin molecules or the average chain length between crosslinks, please? Further explanation in the text would be useful, please.

Author response #19:

Firstly, the percentage of free amino groups in crosslinked nanofibrous samples is obtained and compared to non-crosslinked nanofibers, by ninhydrin assay. Finally, the degree of crosslinking is calculated through Equation (2), as shown in page 5.

This method has well been described in the following reference:

Silva, Simone S., et al. "Novel genipin-cross-linked chitosan/silk fibroin sponges for cartilage engineering strategies." Biomacromolecules 9.10 (2008): 2764-2774

Comment #20:

According to the description in the experimental methods section, up to 5mg/mL of carrageenan was added to a 12% (i.e. 120mg/mL) fibroin solution. That would only reduce the fibroin concentration slightly (to 96% compared with the pure SF material). So, can the authors comment on why the crosslinking degrees were reduced more strongly, please?

Author response #20:

Based on the results, the mechanical properties and degradation are also influenced by adding small amounts of CG to SF. Since these characteristics can be governed by the degree of crosslinking, they can confirm the critical role CG on crosslinking degree.  This may be partly because of the higher fraction of lysine and arginine amino acids in pure FS sample which lead to higher crosslinking degree; however more evaluations are essential for understanding the structural changes.

Comment #21:

Fig. 3c: Is a scale bar missing, please?

Author response #21:

We inserted the missing scale bar in the figure 3c.

Comment #22:

Fig. 4: It is interesting that a relatively small addition of carrageenan (up to 4% w/w) to the silk protein has such a large effect (e.g. ALP activity doubling at 7 days). Could that be due to an uneven distribution of carrageenan within the fibres (e.g. concentrated at the surface)?

It would be interesting to analyse the distribution over fibre cross-sections (e.g. sulphur distribution by SEM with EDX). Can the authors comment, please?

Author response #22:

The authors appreciate your comment! This interesting effect of carrageenan on ALP activity could be attributed to the presence of "sulfonate groups" in k-CG which has also been reported in previous studies (reference 17, 55).

Thank you again for your insightful comment! That's an interesting idea. Unfortunately, at this phase of our project, we didn't get the results of EDX; but we will consider your invaluable comment in our future works.

Comment #23:

P1: '...A functional bone tissue substitutes having...' (suggested: 'Functional bone tissue substitutes having...')

Author response #23:

We have revised the manuscript and your suggestion has been applied.

Comment #24:

P1: 'silica' is not an element (suggested: 'silicon').

Author response #24:

Thank you for mentioning this. We have corrected it in the manuscript.

Comment #25:

P2: '...mainly used because desirable attributes...' (Suggested '...mainly used because of its desirable attributes...')

Author response #25:

According to the respected referee's comment, we have corrected the manuscript.

Comment #26:

P2: '...SF and k-CG polymeric in this study...' (Suggested: '...SF and k-CG polymers in this study...')

Author response #26:

Your suggestion has been applied in the manuscript.

Comment #27:

P3 and elsewhere: The standard method for giving binomial (scientific) names for organisms is that:

(i)the name is given in full and in italics, with the genus capitalised (i.e. shown correctly on P2);

(ii) subsequently, the genus is abbreviated and only the species name is given in full - lower case and in italics)

(iii)e.g. see: https://www.enago.com/academy/how-to-write-scientific-names-in-a-research-paper-animals-plants/

Author response #27:

The authors appreciate your careful comment! According to the respected referee's comment, we used the standard method and rewrote the scientific names given in the manuscript, according to the website guidelines.

Comment #28:

P3 and elsewhere: Strange punctuation of '...and then,..'. Suggested better constructions would be:

'...was boiled in 0.02M Na2CO3 for one hour, then rinsed thoroughly with deionized...'

'...nanofibers were gathered on the aluminum foil surface, then crosslinked...'

etc.

Author response #28:

We replaced the suggested construction, in our revised manuscript.

Comment #29:

P3: '...4 h before being electrospinning to make...' (should be '...4 h before electrospinning to make...')

Author response #29:

We corrected it in the revised manuscript.

Comment #30:

P5: '...a density of 1.0×104 cells/well...' ('4' should be superscripted)

Author response #30:

We superscripted it in the revised manuscript.

Comment #31:

P11: 'spreaded' (should be 'spread)

Author response #31:

We corrected it in the revised manuscript.

The authors would like to appreciate you for your valuable comments and suggestions for improving the paper.

Reviewer 3 Report

The paper describes the possibility to obtain a novel nanofibrous hybrid scaffold based on silk fibroin and different weight ratios of kappa-carrageenan using electrospinning and genipin as crosslinker. The English edition of the manuscript is readable and understandable.

The results are proper for publishing in this journal, however, there are issues that must be solved before its recommendation for publication, such as:

  • Please, in the Introduction, I believe it’s better to replace the UTS values with.. increased by two or even three times in the case of samples obtained with...
  • Please, revise Figue 1.a. all size distribution graphs-with observable separation lines;   SF-GP contact angle image; all images should be uniform in colour and dimensions of the drops.c SG-CG5-GP please fit the broken curve; d. Please, add Y-axis-arbitrary intensity.
  • FTIR/XRD curves-keep the same colours for the samples in all figures; please, improve the resolution.
  • Have you measured the degradation of the samples in protease solution after 30 days or 50 days? It would be usefull to add the datas as bone tissue regenerates after at least 1 month.
  • At Conclusion, please specify what you have brough new in comparison with other studies.

Therefore I would suggest publication of the paper after the major revisions are taken into consideration.

With respect,

Author Response

Reviewer 3:

Comment #1:

The results are proper for publishing in this journal, however, there are issues that must be solved before its recommendation for publication

Author response #1:

We would like to thank the referee for his/her positive feedbacks.

Comment #2:

Please, in the Introduction, I believe it’s better to replace the UTS values with.. increased by two or even three times in the case of samples obtained with...

Author response #2:

According to the respected referee’s concern, we replace the UTS values, in the second paragraph of introduction.

Comment #3:

Please, revise Figure 1.a. all size distribution graphs-with observable separation lines;   SF-GP contact angle image; all images should be uniform in colour and dimensions of the drops.c SG-CG5-GP please fit the broken curve; d. Please, add Y-axis-arbitrary intensity.

Author response #3:

We inserted separation lines in figure 1.a, and uniformed the color in all contact angle images; we also fitted the broken curve in FTIR spectra and added Y-axis-arbitrary intensity in XRD patterns.

Comment #4:

FTIR/XRD curves-keep the same colours for the samples in all figures; please, improve the resolution.

Author response #4:

According to the respected referee’s concern, we used same colors for the samples in all figures and tried to improve the resolution.

Comment #5:

Have you measured the degradation of the samples in protease solution after 30 days or 50 days? It would be useful to add the data as bone tissue regenerates after at least 1 month.

Author response #5:

Thank you for this comment! We replaced our data for degradation and mass loss measurement during one month which are represented at figure 2. (d)

Comment #6:

At Conclusion, please specify what you have brough new in comparison with other studies.

Author response #6:

Thank you very much for this comment! We revised the conclusion and clarified some advantages of our fabricated nanofibrous scaffolds.

Comment #7:

Therefore, I would suggest publication of the paper after the major revisions are taken into consideration.

Author response #7:

We would like to thank the referee for his/her positive feedbacks!

The authors would like to appreciate you for your valuable comments and suggestions for improving the paper.

Round 2

Reviewer 1 Report

The authors responded appropriately to all observations.

Author Response

Reviewer 1:

Comment:

The authors responded appropriately to all observations.

Response:

The authors would like to appreciate you for recommending our manuscript for publication in Biology.

Reviewer 2 Report

Regarding the revised manuscript by Roshanfar et al. I would like to thank the authors for addressing most of my previous comments on their first draft manuscript.

I still doubt the claim (P2) that the main sequence of Bombyx mori silk fibroin (Gly-Ala-Gly-Ala-Gly-Ser)n is hydrophobic.  I believe that is a misunderstanding of the 'hydropathy' index proposed by Kyte and Doolittle (J. Mol. Biol. 1982; 157, 105-132).  More recent work by Makhatadze and Privalov (J. Mol. Biol. 1993, 232, 639-659; J. Mol. Biol. 1993, 232, 660-679) suggests that hydration of most amino acid residues is thermodynamically favoured, predominantly due to the polar peptide groups.  Moreover, this view is supported by a wide range of studies of silk fibroin in native solutions.  Nevertheless, I note that the authors are only repeating what has already been published elsewhere (refs. 10 and 12), so I cannot really object.

Also, I would normally expect the rheology of polymer solutions to be non-Newtonian - in particular, shear thinning.  So, further details of flow history and shear rates would be useful (P3), but I presume that information is not available from the instrument used (Brookfield model LVDVE 230).

In my opinion, however, one significant point remains unanswered.  

On P8 (Fig. 1c), the FTIR trace of the SF fibre specimen appears to show a relatively strong amide II peak (1533 cm-1) and a bimodal amide I peak (relatively weak at 1644 cm-1 and a stronger peak at 1693 cm-1).  While dipolar coupling between peptide groups in antiparallel ß-sheet structures does produce a band around 1690-1700 cm-1, it is usually weak, compared with the main amide I band.  Consequently, in my opinion, there is something unexplained in this spectrum.

I presume the authors have collected a number of spectra from other similar SF samples.  Do they show similar features in the amide I region?

While I cannot see anything unexpected standing out prominently in the other spectra of Fig. 1c, the unexplained feature at 1693cm-1 in the SF spectrum does raise questions concerning sample purity or the proper functioning of the spectrometer. 

In view of that, I am not comfortable recommending publication at this stage, at least in order to allow the authors to review their IR spectra.

Author Response

Reviewer 2:

Comment:
On P8 (Fig. 1c), the FTIR trace of the SF fiber specimen appears to show a relatively strong amide II peak (1533 cm-1) and a bimodal amide I peak (relatively weak at 1644 cm-1 and a stronger peak at 1693 cm-1).  While dipolar coupling between peptide groups in antiparallel ß-sheet structures does produce a band around 1690-1700 cm-1, it is usually weak, compared with the main amide I band.  Consequently, in my opinion, there is something unexplained in this spectrum. I presume the authors have collected a number of spectra from other similar SF samples.  Do they show similar features in the amide I region? While I cannot see anything unexpected standing out prominently in the other spectra of Fig. 1c, the unexplained feature at 1693cm-1 in the SF spectrum does raise questions concerning sample purity or the proper functioning of the spectrometer. 

In view of that, I am not comfortable recommending publication at this stage, at least in order to allow the authors to review their IR spectra.

Author Response:
The authors would like to appreciate you for your comment! Your explanation prompted us to review the FTIR spectrum. You are absolutely right, so we did the FTIR analysis again carefully and present the new results with explanations in figure 1.c.

We noticed that the strong peak in 1600-1700 cm−1 which is related to amide I, was presented at 1673 cm−1 in SF nanofibrous scaffold, so we corrected it in both manuscript's text and figure 1.c. This strong bond showed similar features in previous study (reference 40), which a strong peak was appeared at 1671 cm−1, attributed to amide I.

After GP-crosslinking (SF-GP sample), the bands corresponding to amide I shifted to 1640 cm−1 and a strong amide II peak (1533 cm-1) slightly shifted to 1524 cm−1, which could be due to the change of conformation from random coils or α-helix to β-sheet reminiscent of a more stable SF structure, as reported in previous studies (references 40,41).
The manuscripts provide more information in greater depth regarding the other appeared peaks.

Reviewer 3 Report

Dear Authors,

The manuscript was modified accordingly.

The manuscript is now ready to be published.

Author Response

Reviewer 3:

Comment:
The manuscript was modified accordingly.
The manuscript is now ready to be published.
Response:

The authors would like to appreciate you for recommending our manuscript for publication in Biology.

This manuscript is a resubmission of an earlier submission. The following is a list of the peer review reports and author responses from that submission.